# Numerical Investigation of Performance and Flow Characteristics of a Tunnel Ventilation Axial Fan with Thickness Profile Treatments of NACA Airfoil

**Yong-In Kim [1,2]** , **Sang-Yeol Lee [3]**, **Kyoung-Yong Lee [1]**, **Sang-Ho Yang [3]** and **Young-Seok Choi [1,2,\*]**

1   Thermal & Fluid System R&D Group, Korea Institute of Industrial Technology, Cheonan-si 31056, Korea; yikim89@kitech.re.kr (Y.-I.K.); chrisst@kitech.re.kr (K.-Y.L.)
2   Green Process and Energy System Engineering, Korea University of Science and Technology, Daejeon 34113, Korea
3   Research Department, Samwon Environment & Blower Co., Ltd., 233, Siheung-si 15078, Korea; sang4728@sebco.co.kr (S.-Y.L.); turboh5@hanafos.com (S.-H.Y.)
\*   Correspondence: yschoi@kitech.re.kr; Tel.: +82-41-589-8337

**Abstract:** An axial flow fan, which is applied for ventilation in underground spaces such as tunnels, features a medium–large size, and most of the blades go through the casting process in consideration of mass production and cost. In the casting process, post-work related to roughness treatment is essential, and this is a final operation to determine the thickness profile of an airfoil which is designed from the empirical equation. In this study, the effect of the thickness profile of an airfoil on the performance and aerodynamic characteristics of the axial fan was examined through numerical analysis with the commercial code, ANSYS CFX. In order to conduct the sensitivity analysis on the effect of the maximum thickness position for each span on the performance at the design flow rate, the design of experiments (DOE) method was applied with a full factorial design as an additional attempt. The energy loss near the shroud span was confirmed with a quantified value for the tip leakage flow (TLF) rate through the tip clearance, and the trajectory of the TLF was observed on the two-dimensional (2D) coordinates system. The trajectory of the TLF matched well with the tendency of the calculated angle and correlated with the intensity of the turbulence kinetic energy (TKE) distribution. However, a correlation between the TLF rate and TKE could not be established. Meanwhile, the *Q*-criterion method was applied to specifically initiate the distribution of flow separation and inlet recirculation. The location accompanying the energy loss was mutually confirmed with the axial coordinates. Additionally, the nonuniform blade loading distribution, which was more severe as the maximum thickness position moved toward the leading edge (LE), could be improved significantly as the thickness near the trailing edge (TE) became thinner. The validation for the numerical analysis results was performed through a model-sized experimental test.

**Keywords:** axial fan; thickness; performance; tip leakage flow (TLF); inlet recirculation; National Advisory Committee for Aeronautics (NACA) 3512

## 1. Introduction

An axial flow fan is one of the most important pieces of fluid machinery helping the circulation and ventilation of air in various industrial sites. In particular, the operation of an axial fan in a system where it is difficult to actively circulate air, such as tunnels and underground spaces, is absolutely essential to mankind. In an axial fan, the direction for the suction and discharge of the working fluid (air) is parallel to the shaft, and the blades and guide vanes are paired to form a stage. The blade

raises the total pressure with respect to the working fluid inside an axial fan, and the guide vane is attached as the rear part of the blade to recover the dynamic pressure to static pressure. The pressure acts as a generated energy to transport the working fluid. In some cases, a reverse operation function is installed to control the air or harmful gases under emergencies such as a fire. In cases that require high pressure, additional stages would be added in series as a pair (blade and guide vane). The blades and guide vanes are classified as main components to maximize the performance of an axial fan. As some additional accessories, a hub cap and bell mouth can be added at the front part of an axial fan, and a tail cone or diffusing duct can also be attached at the rear part of an axial fan. Understanding the aerodynamic mechanism of each part is a basic requirement to predict energy consumption, and the rotating part (blade), which is the main focus of this study, is the most important aspect when designing an axial fan.

Due to this necessity, many previous studies on the design of an axial fan were actively conducted, and design techniques were established with empirical coefficients [1]. Moreover, with the recent intensification of air pollution around the world, human society is deeply interested in the supply and exhaust of air through a fluid circulation device, thereby trying to obtain a high quality of life. At the same time, considerations for high efficiency are actively undertaken to prevent the derivation of secondary pollution while minimizing energy consumption. Many previous researchers applied various methods such as the design of experiments (DOE; factorial design method, central composite method, etc.) or optimization techniques (Taguchi method, genetic algorithms, simplex optimization method, free vortex method, response surface method, etc.) to numerical optimization, and successful results were obtained in securing the high efficiency of an axial fan. Bamberger et al. [2] conducted a study to reduce energy loss and noise using the blade sweep angle as a design variable. They adjusted the parameters related to the dimension and streamwise position of maximum camber from the National Advisory Committee for Aeronautics (NACA) profile along the blade span. The aerodynamic optimization was performed with the Nelder–Mead method, and the total efficiency at the design flow rate was selected as the objective function. From their results, it became noticeable that a wave-shaped hub with optimized dimension and streamwise position of maximum camber could prevent unfavorable pressure gradients and wall secondary flow losses. Ren et al. [3] performed an optimization of the performance and noise level using three parameters for the shroud of a small axial cooling fan. The parameters were the distance from the original position in the axial direction (downstream), orifice length at the shroud in the axial direction, and height of the serrated shape. Response surface methodology (RSM) was applied, and the final design of the shroud obtained from the RSM was shown to increase by 21.8% for the volume flow rate and reduce by 3.55 dB for the surface acoustic power level. Lee et al. [4] selected meridional parameters and loading distributions as the initial design variables to lead an optimized design with improved performance through sensitivity analysis. The optimization results were shown to increase by 28.2% for the static pressure with slightly decreased efficiency. In addition, several papers reported on improving the performance for decades [5–9]. On the other hand, most of the previous studies on the performance of axial fans published so far focused on the blade angle or meridional parameters. As a result, reports on other design variables are relatively insufficient. It can be considered that the blade angle and meridional parameters are sensitive to the performance and the stall phenomenon [10–13]. Interest in those design variables is taken for granted to lead to significant improvements in performance.

In addition to the blade angle and meridional design parameters, one of the important design parameters is the tip clearance. In the case of a rotating device that has a casing, tip clearance is essential, thus, previous related studies were performed with various viewpoints. Alavi et al. [14] analyzed the pattern of tip leakage flow (TLF) with changes in the tip clearance size (three sets) and flow rate (two sets). From their results, the intensity of the TLF was proportional to the tip clearance size and inversely proportional to the flow rate, which was analyzed as the cause of deterioration in fan performance directly connected to the energy loss. In addition, the direction of the TLF was changed with respect to the flow rate, and it could be combined with the main flow. Fukano et al. [15] also

confirmed that, as the flow rate changed from high to low, energy loss and noise level were increased with gradual reinforcement of the intensity and velocity of the TLF. As shown by previous studies, the interest in predicting the performance and energy loss using various design variables of an axial fan is very high and still active [16–18].

According to Castegnaro's report [19], which summarized the evolution of aerodynamic design methods for an axial fan in human history, the first step toward controlling air began in 1556 with the aim of ventilating underground tunnels. Continuous efforts have been taken to establish specific design techniques that cover the very wide spectrum of an axial fan. In particular, from the mid-1900s, technical reports were steadily published from military and civil agencies, such as the NACA, and some definitions for an airfoil shape were eventually established. Among the various definitions for an airfoil shape, the axial fan blade of this study is based on the NACA four-digit definition. The meaning of each code in the NACA four-digit representation is shown in Figure 1 for an arbitrary span. The first code is the maximum camber dimension divided by the chord length, which is expressed as a percentage. The second code is the maximum camber position which is defined as the streamwise length from the leading edge (LE) of an airfoil on the chord line, where the LE and trailing edge (TE) should be normalized to 0 and 10, respectively. The remaining third and fourth codes are defined as the maximum thickness dimension of an airfoil, which is also expressed as a percentage to the chord length. In the four-digit representation, however, the thickness profile including the maximum position is not defined; instead, it is determined using an empirical formula (Equation (1)) with the specific coefficients.

$$y = \frac{\delta_m}{c}\left(1.4845\sqrt{x} - 0.63x - 1.758x^2 + 1.4215x^3 - 0.5075x^4\right) \tag{1}$$

where $\delta_m$, $c$, $x$, and $y$ are the maximum thickness dimension of the airfoil, chord length of the airfoil, normalized streamwise position of the camber line, and actual thickness dimension of the airfoil (which can be replaced with $\delta$ in this study), respectively. According to this empirical equation, the thickness profile of an airfoil represents the specific distribution with a maximum thickness position at 30% from the LE in any case. Airfoils having a specific distribution of the thickness profile can secure good performance and noise level compared to airfoils with a linear distribution [1]. This is a design technique based on empirically accumulated functions, and high performance can generally be expected. However, verification of the empirical equation is essential in the design of an axial fan because the blade universally has a twisted stacking shape, and specific phenomena or local energy losses are difficult to predict when the maximum thickness position is changed in some way. There is also uncertainty when making an assumption that the casting or post-work related to the roughness treatment is performed accurately during the manufacturing process for an airfoil. Considering the situation for the manufacturing site where most steps are a function of the technique and concentration of the employee, even when conducted under an automation system, its importance should be fully evaluated. The effect of the maximum thickness dimension of an airfoil (referred to as the last two codes of the NACA four-digit representation) on the performance of an axial fan was studied by Sarraf et al. [20]. In their study, the number of sets for the maximum thickness dimension was two for 3 and 10 mm. The design specification of the axial fan was 270 Pa, 2650 m$^3\cdot$h$^{-1}$, and 260 rad$\cdot$s$^{-1}$ for the pressure, volume flow rate, and rotational speed, respectively. From the results, it was confirmed that, when the thickness was 10 mm, the pressure decreased at the design flow rate and the best efficiency point (BEP) moved to the region of a lower flow rate.

In this study, the effect of the maximum thickness position of an airfoil on the performance and aerodynamic characteristics of an axial fan was examined through numerical analysis. Only the maximum thickness position was selected as a variable, and all other design specifications and parameters (blade angle, chord length, meridional plane, flow rate, rotational speed, etc.) were unaltered. That led to one-factor analysis for the streamwise position of maximum thickness. The airfoil was based on the NACA 3512. The effect of the maximum thickness position was analyzed from various perspectives such as the performance of the axial fan, sensitivity analysis for each span,

turbulence kinetic energy (TKE) distribution, trajectory of the TLF, inlet recirculation, and correlation between the energy loss and TLF coefficient. The internal flow phenomena were thoroughly discussed in terms of fluid mechanics. Detailed attempts were made to visualize the internal flow from the results of numerical analysis, which were interrelated. The blade loading distribution is also presented with respect to the thickness profile. Moreover, an additional discussion related to the problem of the set showing an unstable loading distribution is presented with the adjustment of the thickness near the TE. The validation for numerical analysis was performed through a model-sized experimental test.

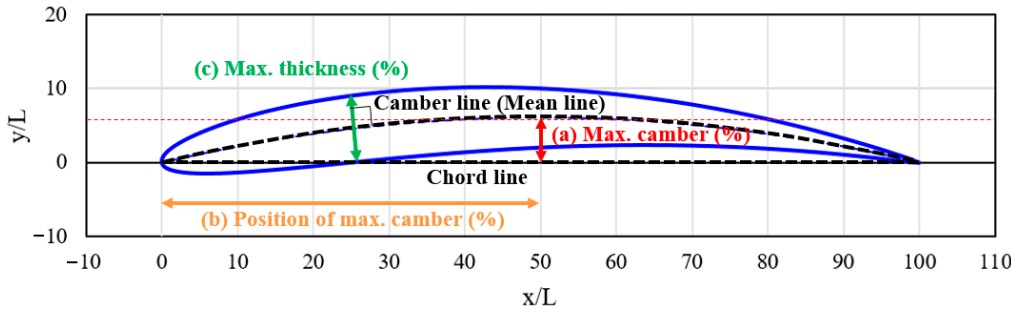

**Figure 1.** Definition of National Advisory Committee for Aeronautics (NACA) four-digit airfoil.

## 2. Axial Fan Model

### 2.1. Design Specification

The axial fan of this study was designed for ventilation in underground spaces and tunnels as a 75 kW·h$^{-1}$ energy-consuming device. As shown in Figure 2, it was a one-stage axial fan with a pair of rotor (10 blades) and stator (11 guide vanes). A hub cap and bell mouth were attached to the front part of the axial fan. A silencer could have been attached to the front or rear part of the axial fan in consideration of the installed space [21], but it was not considered in this study. The blades were designed as a three-dimensional (3D) shape with the NACA 3512 airfoil for all spans. The guide vanes were designed as a two-dimensional (2D) geometry, resulting in the same sectional shape in the spanwise direction. Both blade and guide vane had a shorter meridional and chord length at the shroud span than the hub span. The tip clearance ($\delta_t$) was placed at 0.99–1.00 on the basis of the normalized span ($r^*$). Table 1 lists the details of the design specifications of the reference model. The specific speed ($N_s$), flow coefficient ($\Phi$), and pressure coefficient ($C_p$) were nondimensionalized with Equations (2)–(4), respectively.

$$N_s = \frac{\omega_N \sqrt{Q}}{(P_t/\rho)^{\frac{3}{4}}} \tag{2}$$

$$\Phi = \frac{C_{m2}}{U_2} \tag{3}$$

$$C_p = \frac{P_t}{\frac{1}{2}\rho W_1{}^2} \tag{4}$$

where $\omega_N$, $Q$, $P_t$, $\rho$, $C_{m2}$, $U_2$, and $W_1$ denote the angular velocity, volume flow rate, total pressure, air density, meridional component of absolute velocity at the blade outlet, circumferential component of rotational velocity at the blade outlet, and relative velocity of air at the blade inlet, respectively.

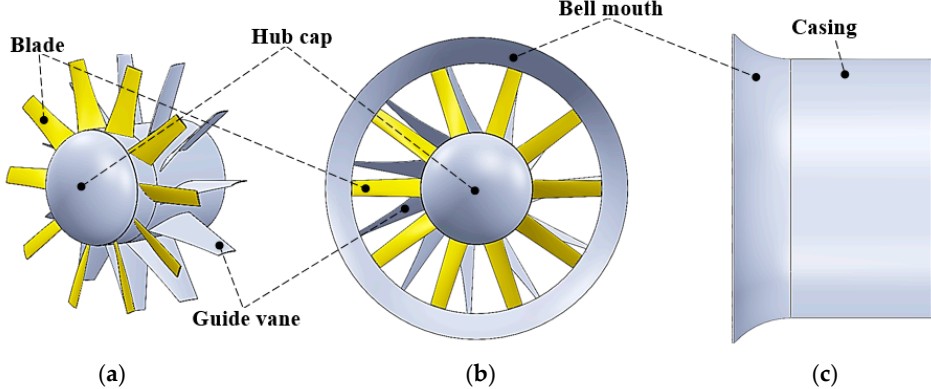

**Figure 2.** Axial fan unit: (**a**) three-dimensional (3D) view; (**b**) front view; (**c**) side view.

**Table 1.** Design specifications and parameters of the reference axial fan.

| Parameter | Value | Unit |
|---|---|---|
| Specific speed ($N_s$) | 7.77 | (-) |
| Flow coefficient ($\Phi$) | 0.29 | (-) |
| Pressure coefficient ($C_p$) | 0.12 | (-) |
| Rotational speed ($N$) | 1185 | (rpm) |
| Hub ratio ($r_h/r_s$) | 0.44 | (-) |
| Tip clearance ratio ($\delta_t/r_s$) | 0.0056 | (-) |
| Solidity ($c/s$) | 0.769 (hub), 0.155 (shroud) | (-) |
| Setting angle [1] | 49.7 (hub), 23.1 (shroud) | (degree) |
| No. of blades/guide vanes | 10/11 | (ea) |
| Airfoil series | NACA 3512 | (-) |

[1] Tangential definition.

## 2.2. Thickness Profile

Figure 3a shows the thickness profile of each span under the adjustment of the maximum thickness position every 5% in the streamwise direction. The maximum thickness position was expressed as a percentage of the normalized value of 0 and 1 for the LE and TE, respectively. A total of eight sets were designed in the range of 15–50%. The maximum thickness position of each set was designed to be distributed at the same position over the entire span. The 30% set was designed to have a distribution that followed the empirical formula in Equation (1). As a special feature, the thickness of the LE and TE was increased to 3 mm to consider the casting and manufacturing processes. The sectional shapes of the hub and shroud span with the different thickness profiles of Figure 3a are shown in Figure 4a,b. Each shape is displayed while maintaining the actual scale. Figure 3b shows the thickness profile under adjustment near the TE. This is the perspective of a follow-up discussion to present a solution to the set with a critical problem among the sets in Figure 3a. There were three sets for thickness treatments near the TE, and the sectional shapes of each span are shown in Figure 4c,d. Here, the definition of the thickness in this study was the normal direction to the camber line, not the tangential direction.

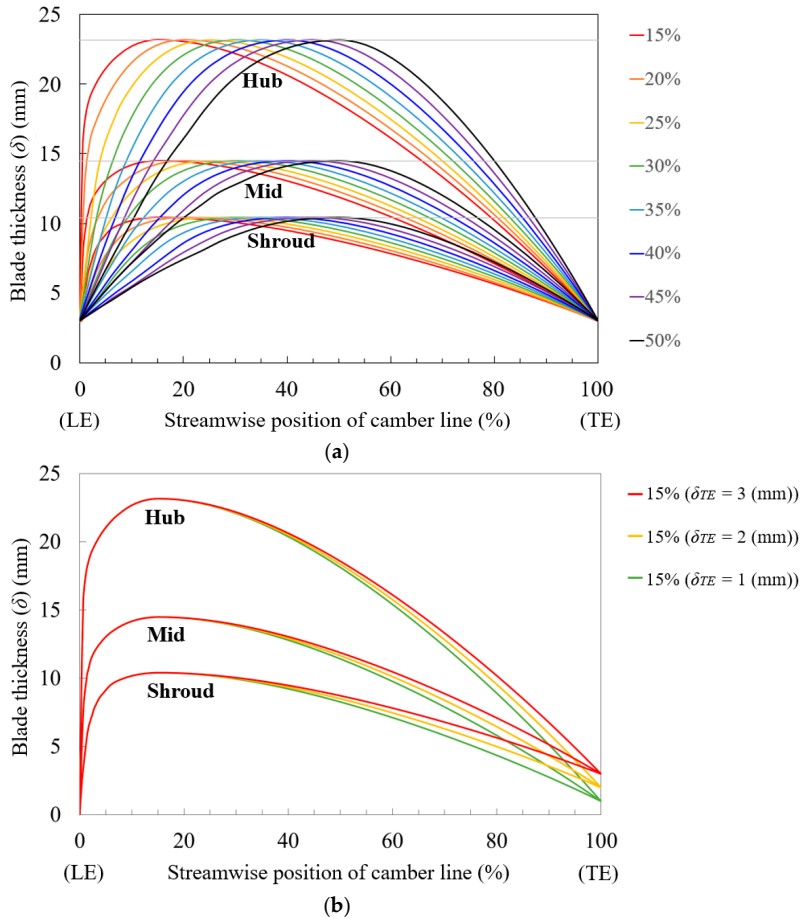

**Figure 3.** Thickness profiles of airfoil with variation in (**a**) maximum thickness position and (**b**) trailing edge thickness treatment.

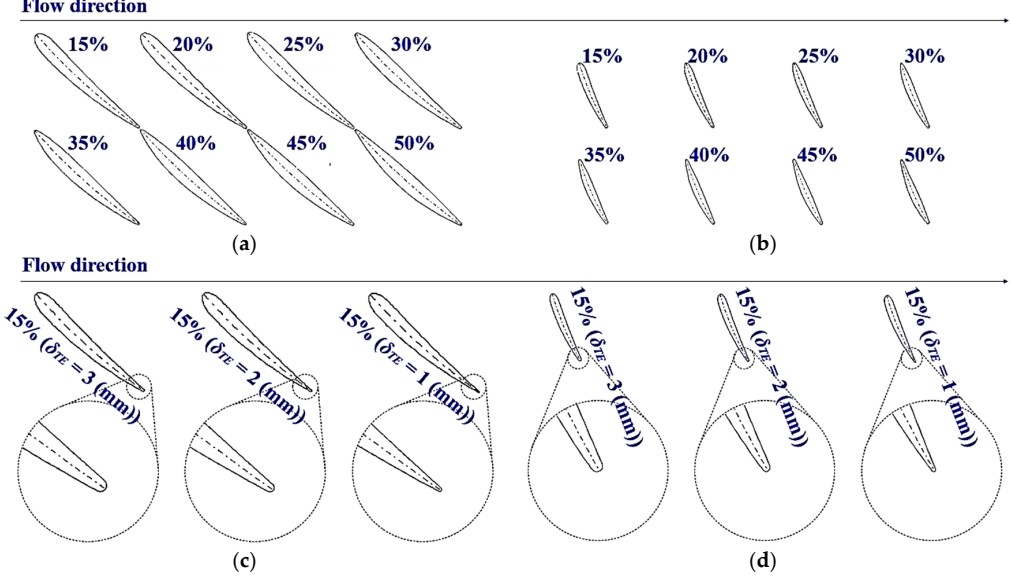

**Figure 4.** Sectional shapes of the airfoil for the maximum thickness position at the (**a**) hub and (**b**) shroud span, and the thickness treatment near the trailing edge at the (**c**) hub and (**d**) shroud span.

## 3. Numerical Analysis Set-Up

### 3.1. Computational Domain and Grid System

Figure 5 shows the whole flow passage and grid system for numerical analysis. As shown in Figure 5a, the computational domain included only the rotating part. It was assumed that the guide vane would be designed with a good understanding of the flow pattern at the outlet of the blade; thus, the guide vane was not considered in the computational domain to highlight the focus on the blade. The hub cap and bell mouth which could be attached in front of the blade were also not considered in the computational domain because they had almost the same effect as an extended straight passage [22]. In the case of a hub cap, if a round or ellipse shape is approximately secured except for the certain blunt shape, the performance shows nearly the same as that of the extended straight passage. The attachment of a hub cap affects the pressure rise, blade loading, and limiting streamline distribution below about a 25% span, but has little effect above a 25% span [23]. In the case of a bell mouth, if the axial distance at the shroud from the blade LE to the bell mouth is tight, it can significantly affect the inlet flow and the TLF, which directly correlates with the performance [24–26]. However, in order to stabilize the inlet flow near the shroud sufficiently [27], the axial distance of the axial fan in this study was secured to be at least 0.4 times the radius of shroud. Although the effects of the hub cap and bell mouth could not be completely excluded, the axial fan for the experimental test was designed to minimize these effects, as validated in the section below. Moreover, in order to escape these effects and concentrate on the effect of the maximum thickness position and TE thickness treatment on the blade, the numerical domain for all sets in this study was prepared only for the blade with an extended straight passage. Meanwhile, periodic conditions were implemented to reduce the numerical time; thus, the single passage with one blade was selected with a pitch angle of 36°.

The grid system was generated using ANSYS CFX TurboGrid 16.1 with a hexahedral type. The grid system near the inlet surface and LE are enlarged in Figure 5a. That of the blade tip are shown in Figure 5b. Here, $r^*$ is a normalized value obtained from the hub to the shroud span as 0 and 1. To secure a high quality of the grid system, $y+$ was kept below 35 considering the stable aspect ratio. The automatic wall function with a nonslip and smooth wall condition was assigned to avoid the effect of $y+$ on the numerical results [28–30]. Under the same topology factor of $y+$, the grid test was conducted with the results shown in Figure 5c. The efficiency in Figure 5c was nondimensionalized with that of the selected grid system. The grid system which does not affect the performance of an axial fan was selected through the grid test, and a grid system with 810,000 nodes was employed for the numerical analysis.

### 3.2. Governing Equation and Boundary Condition

A commercial software, ANSYS CFX 16.1, was used for the numerical simulation. As a governing equation, the Reynolds-averaged Navier–Stokes (RANS) equations were applied in the 3D and steady-state conditions. The equations are based on the conservation laws for mass, momentum, and energy; however, this study focused on the isothermal condition, allowing the equation for the conservation of energy to be ignored. The equations for the conservation of mass (Equation (5)) and momentum (Equation (6)) can be stated as follows:

$$\frac{\partial \rho}{\partial t} + \frac{\partial (\rho u_i)}{\partial x_i} = 0 \tag{5}$$

$$\frac{\partial (\rho u_i)}{\partial t} = -\frac{\partial p}{\partial x_i} + \frac{\partial}{\partial x_j}\left[\mu\left(\frac{\partial u_i}{\partial x_j} + \frac{\partial u_j}{\partial x_i} - \frac{2}{3}\frac{\partial u_r}{\partial x_r}\delta_{ij}\right)\right] + \rho F_i \tag{6}$$

where $t$, $x$, $u$, $p$, and $F_i$ denote the time, coordinate (substituted as $y$ or $z$), velocity (substituted as $v$ or $w$), pressure, and body force, respectively. The terms in square brackets in Equation (6) denote the viscous stress tensor, $\tau_{ij}$. For the first term in each equation, there is no change in density over time because it

is under incompressible conditions. The maximum Mach number near the shroud (local) in the flow passage is 0.42 at 25 °C, which could be regarded as a subsonic flow. In general, subsonic flow can be classified as an incompressible flow (Mach number < 0.3) with no change in density and compressible flow (Mach number < 0.8) having a slight gradient for the density without any shock waves. When the Mach number becomes larger than 0.8, it has to be treated definitely as a compressible flow having a remarkable gradient for the density with shock waves. The validation for the incompressible condition in this numerical study was performed using the experimental test described in the next section. Meanwhile, a high-resolution discretization with a second-order approximation was applied to ensure minimized numerical convergence, instead of first-order schemes [31–33]. The root mean square (RMS) residuals of the governing equations for mass and momentum were kept below $1.0 \times 10^{-5}$.

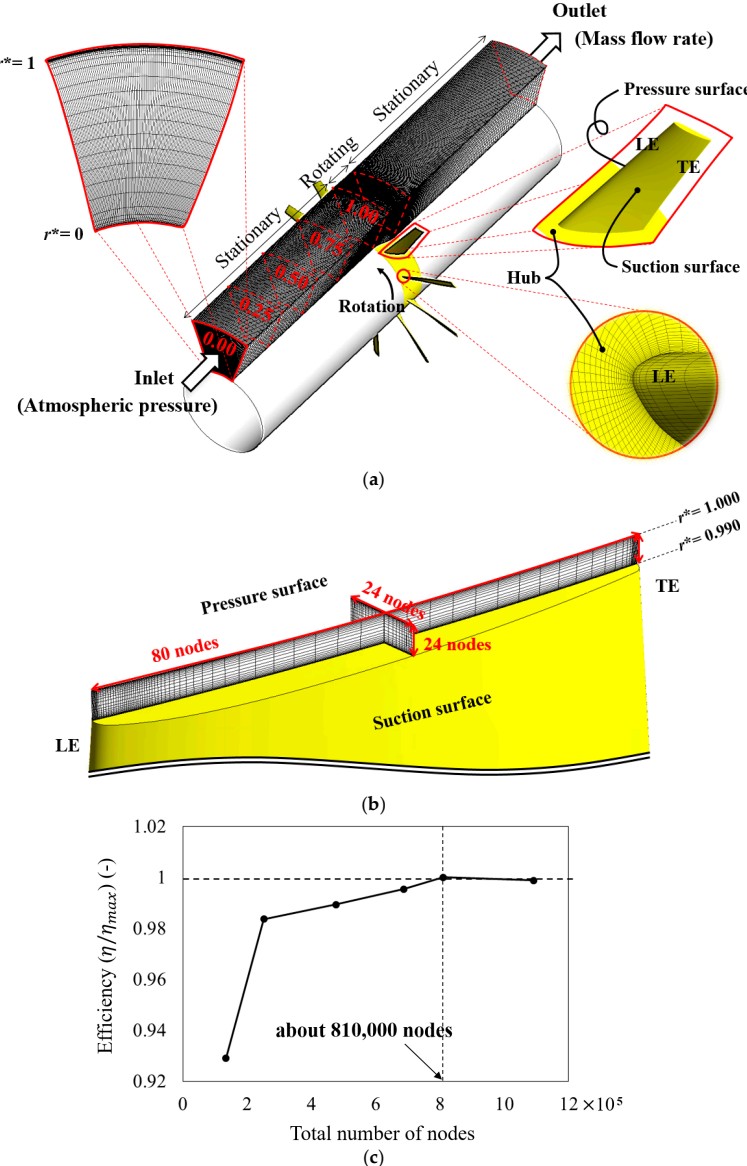

**Figure 5.** Computational domain and grid system: (**a**) flow passage; (**b**) tip clearance; (**c**) result of grid test.

The turbulence model was applied as the shear stress transport (SST) model according to $k$-$\omega$. As the inlet boundary condition, the level of turbulence intensity was selected as medium, which corresponds to the range of 1–5%. Medium turbulence intensity is recommended for flow in not-so-complex devices such as large pipes, fans, wind tunnels, or ventilation flows [34]. Here, a higher level of turbulence

intensity could be applied to high-speed flow inside complex geometries such as heat exchangers and rotating turbomachinery (turbines and compressors). Figure 6 shows the calculated turbulence intensity and turbulence length scale which were averaged in the circumferential direction on each plane indicated in Figure 5a. Each plane was placed with equal intervals from the inlet (0.00) to the outlet (1.00) of the inlet domain (stationary), and the data were specified to the 25% set for the maximum thickness position, as an example. The equations for turbulence intensity ($T_u$), turbulence kinetic energy (TKE; $k$), turbulence length scale ($l$), and turbulence eddy frequency ($\omega$) are as follows:

$$T_u = \sqrt{\frac{\frac{1}{3}\left[(u-\overline{u})^2 + (v-\overline{v})^2 + (w-\overline{w})^2\right]}{\left(\overline{u}^2 + \overline{v}^2 + \overline{w}^2\right)}} = \sqrt{\frac{\frac{2}{3}k}{\left(\overline{u}^2 + \overline{v}^2 + \overline{w}^2\right)}} \tag{7a}$$

$$k = \frac{1}{2}\left[(u-\overline{u})^2 + (v-\overline{v})^2 + \left(w-\overline{w}\right)^2\right] \tag{7b}$$

$$l = \frac{k^{3/2}}{kC_\mu\omega} \tag{8a}$$

$$\omega = \frac{k}{\nu(\mu_T/\mu)} \tag{8b}$$

where $u$, $v$, $w$, $\overline{u}$, $\overline{v}$, and $\overline{w}$ denote the instantaneous velocity and the averaged velocity in the $x$-, $y$-, and $z$-direction in an orthogonal coordinate system, respectively. $C_\mu$, $\nu$, and $\mu_T/\mu$ also denote the nondimensional model constant (0.09), kinematic viscosity, and eddy viscosity ratio. In this study, the $k$-$\omega$-based SST model was applied, allowing the turbulence length scale to be calculated from $\omega$. Meanwhile, the flow direction from the inlet boundary was set as normal, and the options of atmospheric pressure and mass flow rate were given to the inlet and outlet boundary conditions, respectively. The frozen rotor method was applied to the interfaces of the rotating part, because there was no stator at the front and rear of the rotor. This made it possible to secure more details for the flow field without any average. The working fluid was air at 25 °C. Finally, the parallel computations were run on personal computers (PCs) with an Intel® Xeon® central processing unit (CPU) X5690, clocked at 3.47 GHz with a dual processor. The numerical time spent per each set was approximately 1 h.

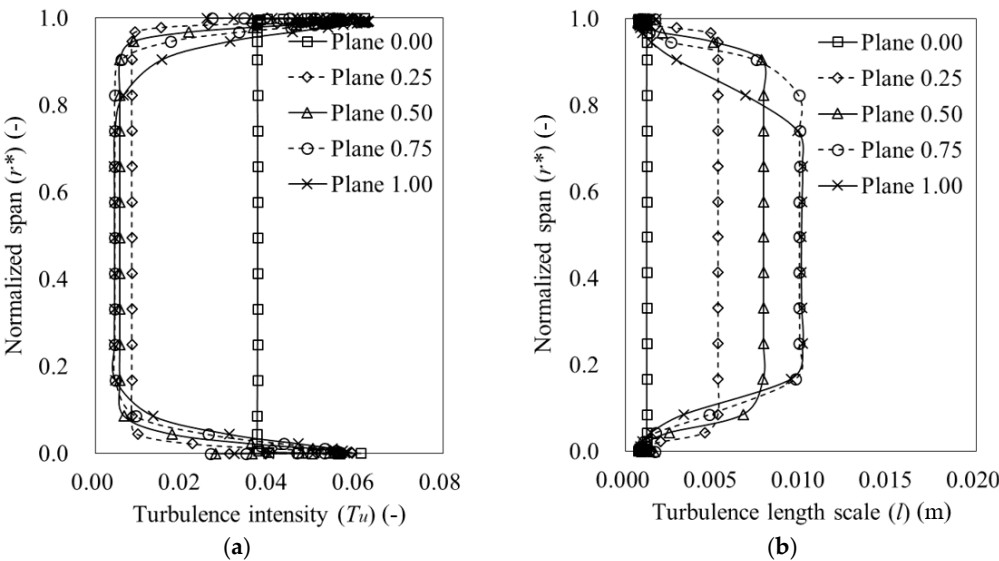

**Figure 6.** (**a**) Turbulence intensity and (**b**) turbulence length scale from hub to shroud span.

## 4. Result and Discussion

### 4.1. Validation for Numerical Analysis

The results of the numerical analysis were validated through an experimental test. The whole process and the facilities for the experimental test were fully implemented in accordance with the international standard, ANSI/AMCA 210-07 [35]. As the experimental test for the prototype and the validation of all the sets in this study were not practical, a small-scaled model test was conducted only for the 25% set for maximum thickness position in accordance with the similarity law. Figure 7a shows the small-scaled rotor that was about 0.23 times smaller than the prototype. It was manufactured a perfect scale ratio including the tip clearance and precise edge treatment using five-axis machining technology. Figure 7b is a photograph of the small-scaled model connected to the fan test facility at the Korea Institute of Industrial Technology (KITECH). The small-scaled model was an assembly including a bell mouth, hub cap, blade, and guide vane, as shown in Figure 2. The 0.5 HP (0.373 kW·h$^{-1}$) class motor with three-phase electric conversion was selected to meet the estimated limit on the basis of the similarity law and was placed inside the hub duct. The schematic diagram of the experimental test facility is shown in Figure 7c. The experimental type of the outlet chamber set-up was adopted among various methods for the fan test. The outlet of the axial fan was connected to the inlet of the chamber with a straight duct having twice the length of the diameter of the axial fan. The flow settling means were installed inside the chamber in accordance with the proper porosity [36]. The density was calculated from the measurement of dry-bulb temperature, barometric pressure, and relative humidity in the laboratory. The average measurements during the test for dry-bulb temperature at the front of the nozzles inside the chamber, dry-bulb temperature at the rear of the nozzles inside the chamber, dry-bulb temperature in the laboratory, barometric pressure in the laboratory, and relative humidity in the laboratory were 24.24 °C, 24.08 °C, 24.92 °C, 101,100.39 Pa, and 46.01%, respectively. The flow rate was controlled with eight nozzles inside the chamber and was calculated using the differential pressure of the nozzles. The differential pressure was measured with pressure manometers. The rotational speed was obtained and verified with a stroboscope and laser tachometer. The range of the flow rate out of the measuring points was controlled with a servo blower behind the chamber. The data acquisition (DAQ) system was incorporated with an averaging function for all outputs (measured data) for 6 s. All instruments in the test facility were verified for annual certification, such that the uncertainty of measurement for the dry-bulb temperature detectors, pressure manometers, and stroboscope was 0.07 °C for the measuring range of 0–60 °C, 0.001–0.005 kPa for the measuring range of 0–1.33 kPa, and 0.1–1 rpm for the measuring range of 40–35,000 rpm, respectively.

Figure 8 shows the validation between the numerical analysis and experimental test. The data of each axis were nondimensionalized with the value of each design flow rate. The experimental data were presented at 0.1 intervals as the nondimensionalized flow rate with 10 repeated measurements. Before calculating the pressure (nondimensionalized as $C_p$), the density and rotational speed from the experimental test results were converted to the same values with the boundary condition of the numerical analysis. From the validation results, the tendency of received energy in terms of the working fluid, which could be substituted as the pressure rise from the axial fan, tended to be very similar in both the experimental test and the numerical analysis. Each curve showed almost the same slope including the off-design flow rates ($\Phi/\Phi_{des} \approx 0.8$–1.1), while it deviated somewhat at the high flow rate ($\Phi/\Phi_{des} \approx 1.2$). In the experimental test, all components were considered as full assembly, whereas, in numerical analysis, only the rotating part (blade) was considered. Nevertheless, the coincidence of each slope means that the components, except the blade, were designed with a good understanding of aerodynamics to have little effect on the performance of the axial fan. Meanwhile, the expanded uncertainty, including the influence of scattered data from repeated measurements (type A) and the uncertainty of measuring instruments (type B), is indicated as black bars in the enlarged graph of Figure 8. The expanded uncertainties at each flow rate of $\Phi/\Phi_{des} \approx 0.8$, 0.9, 1.0, 1.1, and 1.2 were only 0.004, 0.004, 0.006, 0.007, and 0.005 on the *y*-axis, where the level of confidence and the coverage factor

were 95% and 2, respectively. On the other hand, a sudden dissipation with a positive slope can be seen in the range of $\Phi/\Phi_{des} \approx 0.6–0.8$. This phenomenon is referred to as "stall" caused by an increase in incidence angle due to a decrease in flow rate. When an axial fan enters the stall region during operation, significant energy loss, vibration, and noise are generally caused [37,38]; thus, at least at the designed flow rate, the stall region should be avoided from the viewpoint of saving energy and increasing lifetime for an axial fan. The axial fan of this study was not related to the stall region near the design flow rate.

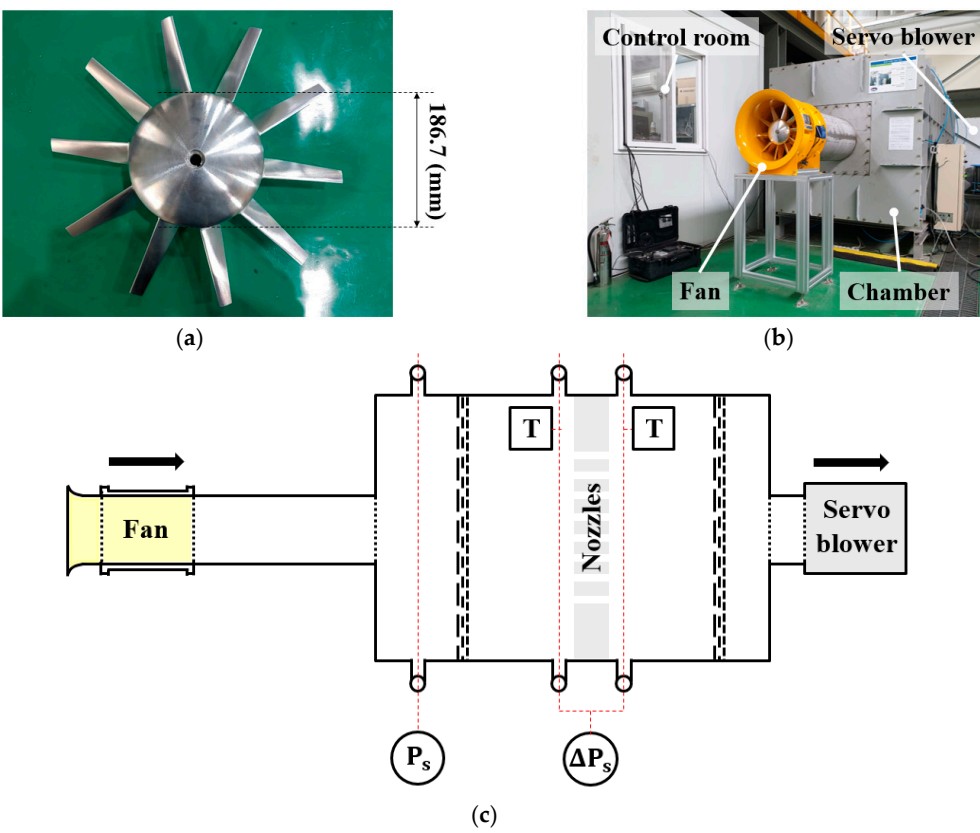

**Figure 7.** Experimental test set-up: (**a**) photograph of the small-scaled rotor; (**b**) photograph of the small-scaled model fan and test facility; (**c**) schematic diagram of test facility.

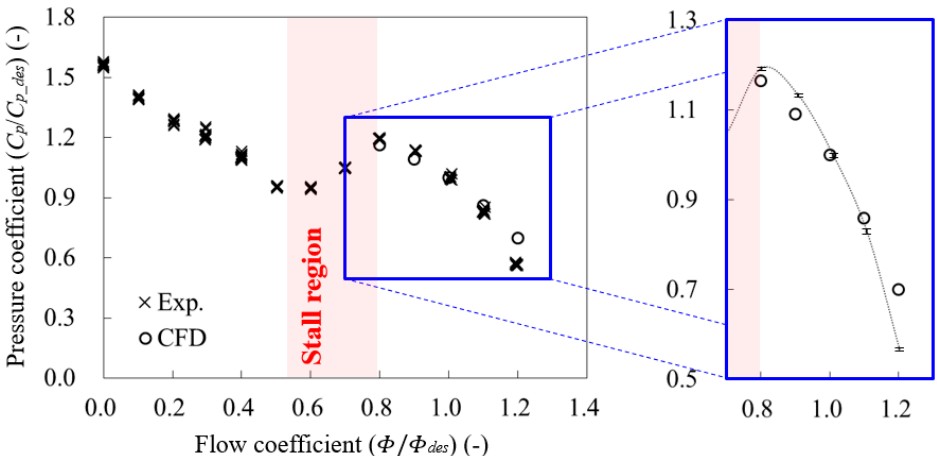

**Figure 8.** Validation between the numerical and experimental results with expanded uncertainty analysis.

## 4.2. Aerodynamic Performance

Figures 9 and 10 are presented to analyze the effect of the maximum thickness position of an airfoil on the performance of the axial fan. The performance is discussed in terms of total pressure rise and efficiency. Each set was designed to be distributed at the same position of maximum thickness over the entire span. Figure 9 shows the performance curves including the off-design flow rates, and Figure 10 compares the performance only for the design flow rate. As a reference, the data of the 25% set in Figure 9a were the same as those in Figure 8. The first conclusion from Figure 9 is that the maximum thickness position affected the performance of the axial fan. Both performance curves for the total pressure and efficiency showed the up–down shifts while maintaining the slope with a difference in the maximum thickness position. In particular, as shown in Figure 9a, the total pressure rise decreased as the maximum thickness position moved toward the TE in the whole range including the off-design flow rates ($\Phi$ = 0.2285–0.3425). In the 15% set, however, the slope tended to be horizontal in the range of low flow rate, away from the other sets. The 20% and 25% sets also started to be deformed at $\Phi$ = 0.2285. This total pressure drop in the low flow rate implies that the stall margin can be decreased as an axial fan enters the stall region, as confirmed in the above validation section. On the other hand, the efficiency was predicted to be generally high in the 30% and 35% sets as shown in Figure 9b. Left–right shifts of the best efficiency point (BEP) were not observed, but a sharp drop in efficiency was confirmed at the low flow rate ($\Phi$ = 0.2285) for the 15%, 20%, and 25% sets, showing the same tendency as the total pressure rise. The most significant drop in efficiency was observed in the 15% set, where the inflection was most severe. Meanwhile, as presented in Figure 10 for the design flow rate, it can be confirmed that the total pressure decreased as the maximum thickness position moved toward the TE. In terms of efficiency, generally high performance could be expected when the maximum thickness position was in the range of 20–35%, including for the 30% set. The efficiency was highest for the 30% set, which was established using the empirical formula in Equation (1). Although the 15% set at the design flow rate showed the highest pressure rise, it failed to ensure efficiency.

In order to conduct a sensitivity analysis on the effect of the maximum thickness position for each span on the performance at the design flow rate, the design of experiments (DOE) method was applied with a full factorial design. A total of 16 sets were generated as listed in Table 2 through a full factorial design for the two factors of the hub and shroud span. The target range of the maximum thickness position constituted the 20%, 25%, 30%, and 35% sets, which were predicted to perform with high performance.

**Table 2.** Maximum thickness position and performance of each sensitivity analysis set.

| Set No. | Maximum Thickness Position (%) | | Performance | |
|---|---|---|---|---|
| | Hub | Shroud | $C_p$ (-) | $\eta/\eta_{ref}$ (-) |
| 1 | 20 | 20 | 0.122 | 0.998 |
| 2 | 25 | 20 | 0.117 | 0.998 |
| 3 | 30 | 20 | 0.117 | 0.999 |
| 4 | 35 | 20 | 0.115 | 0.998 |
| 5 | 20 | 25 | 0.119 | 0.995 |
| *6* | *25* | *25* | *0.121* | *1.000* |
| 7 | 30 | 25 | 0.117 | 0.999 |
| 8 | 35 | 25 | 0.115 | 0.998 |
| 9 | 20 | 30 | 0.118 | 0.995 |
| 10 | 25 | 30 | 0.117 | 0.999 |
| 11 | 30 | 30 | 0.119 | 1.001 |
| 12 | 35 | 30 | 0.113 | 0.996 |
| 13 | 20 | 35 | 0.117 | 0.995 |
| 14 | 25 | 35 | 0.116 | 0.998 |
| 15 | 30 | 35 | 0.115 | 0.998 |
| 16 | 35 | 35 | 0.116 | 0.998 |

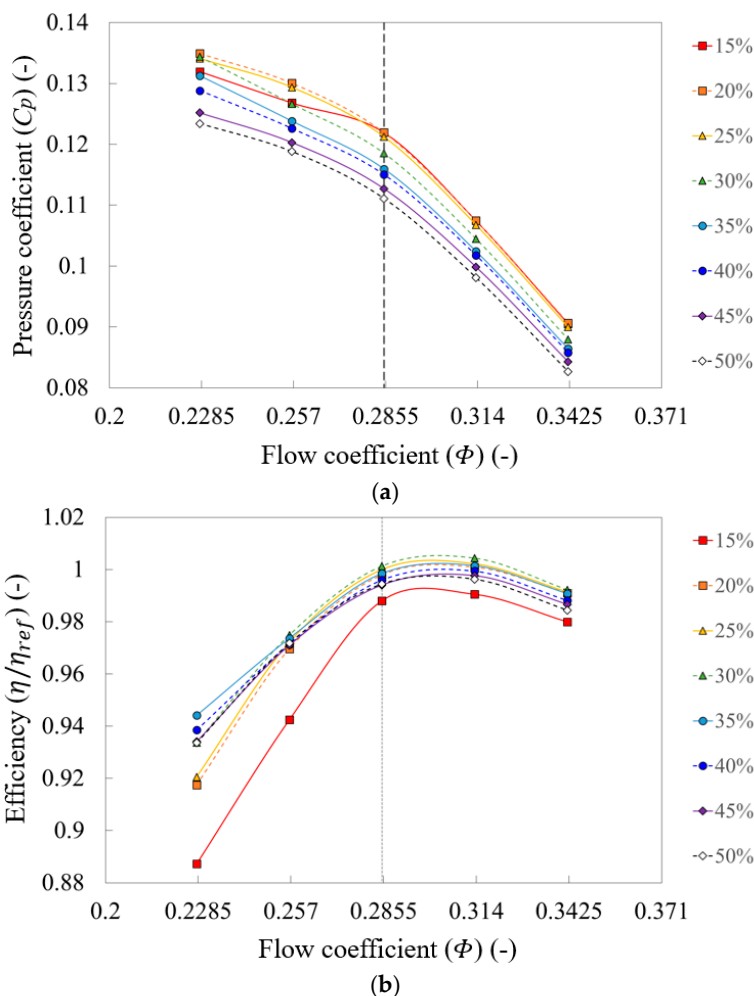

**Figure 9.** Performance curves with different maximum thickness positions ($\Phi/\Phi_{des} = 0.8$–$1.2$): (**a**) total pressure; (**b**) efficiency.

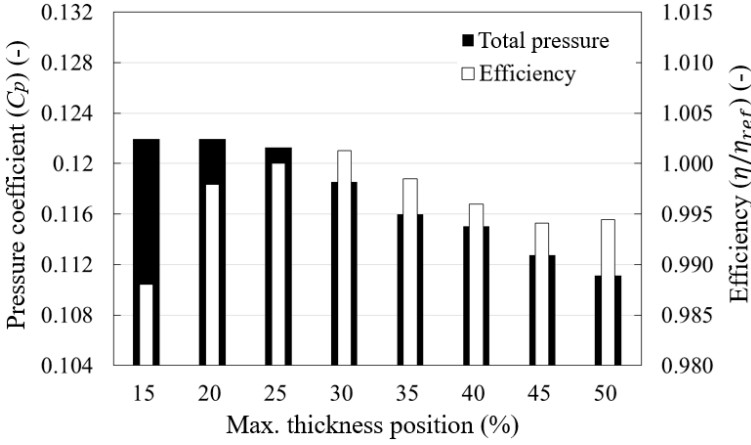

**Figure 10.** Performance with different maximum thickness positions ($\Phi/\Phi_{des} = 1.0$).

The main effect plots are presented in Figure 11 for the sensitivity analysis results. According to the results confirmed in Figure 10, the total pressure tended to decrease in both the hub and shroud span as the maximum thickness position moved toward the TE. In particular, there was a greater impact near the hub span with a steeper slope than the shroud span. Meanwhile, the effect of the maximum thickness position on the efficiency showed a remarkable difference between the hub and shroud span. In the hub span, the highest efficiency could be expected for the 30% set, showing the same tendency as in Figure 10. On the other hand, in the shroud span, the efficiency tended to decrease as the maximum thickness position moved toward the TE. Thus, for the axial fan in this study, the internal flow mechanism near the hub span could mainly determine the overall performance, while that of the shroud span obtained the unidentified phenomena. This remarkable difference in the main effect plots is described in the sections below in terms of the TKE and leakage loss.

*4.3. Internal Flow Characteristic*

In this section, the effect of the maximum thickness position of an airfoil on the performance and aerodynamic phenomena at the design flow rate is presented from various perspectives. As a reminder, the maximum thickness position of each set was distributed with the same percentage over the entire span.

Figure 12 shows the turbulence kinetic energy (TKE; $k$) distribution on each span in a rotating domain. The observation planes and legend are shown in Figure 12i. The legend allows for the entire span in the same range; however, the lowest limit for the hub span ($r^* = 0.01$) was set to 12 to avoid a distribution in the boundary layer and focus on the core region. The TKE was defined with the sum of vectors in three directions, as denoted in Equation (7b). This is a well-known analytical technique to predict the distribution and intensity of turbulence in a fluid passage [39–41], and it tends to agree well with the vector distribution of actual turbulence from the experimental data using particle image velocimetry (PIV) [42]. It can be seen that the TKE near the hub was mainly distributed as a separation of the blade suction surface (SS) for all sets in Figure 12a–h. However, the TKE-occupied region over the upper limit of the legend, as indicated by the red color, became narrower for the 25%, 30%, and 35% sets of Figure 12c–e. In particular, the narrowest region was observed for the 30% set, which performed with the highest efficiency, as shown in Figure 10. As the TKE-occupied region near the hub span was narrow and weak, the efficiency of the axial fan increased, coinciding with the tendency of the main effect plot for the hub span in Figure 11b. Therefore, in this study, a correlation could be established between the TKE-occupied region near the hub span due to the maximum thickness position and the efficiency of the axial fan. On the other hand, a band-shaped distribution of the TKE was observed near the shroud span in the rotational direction. This band-shaped TKE, which could be assumed as the tip leakage flow (TLF), is re-presented using a blade-to-blade view in Figure 13. As the percentage of the maximum thickness position increased, the intensity and occupied region of the TKE tended to decrease, as indicated by the white dotted line. This tendency contradicted the result of the main effect plot for the shroud span in Figure 11b. A detailed analysis of the energy loss due to the TLF with respect to this contradictory tendency continues below. Meanwhile, the TKE was distributed as a wake near the blade TE in all spans, including $r^* = 0.25$, 0.50, and 0.75. Its distribution and intensity were hardly affected by the variation in maximum thickness position.

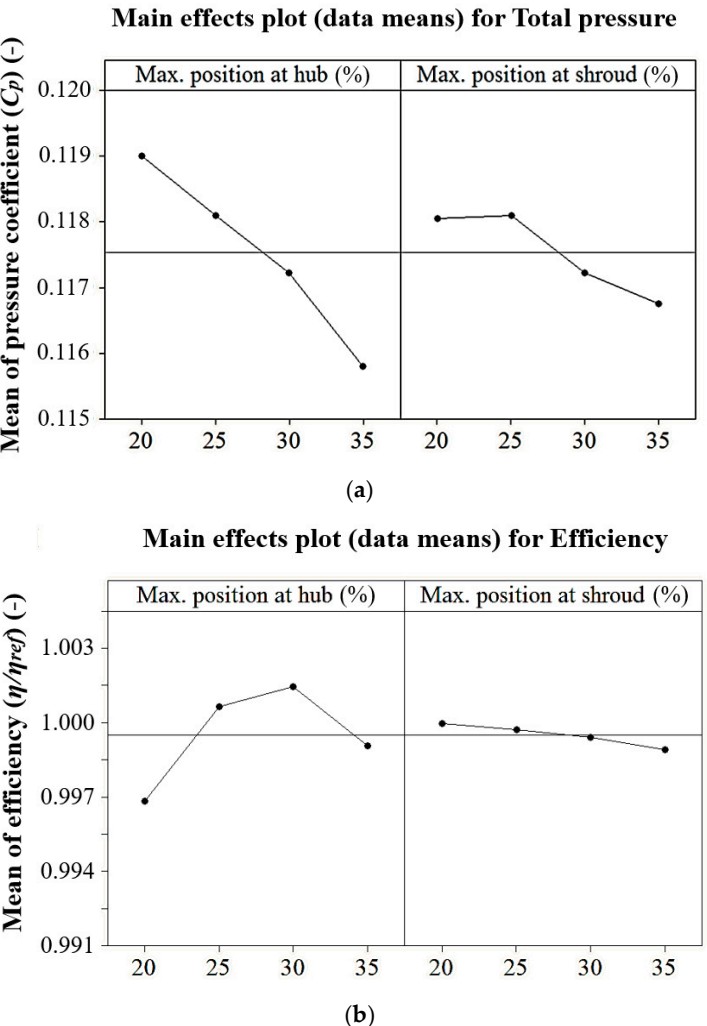

**Figure 11.** Main effect plots for the maximum thickness position of each span: (**a**) total pressure; (**b**) efficiency.

Figure 14 shows the internal flow field near the shroud span with the limiting streamlines and *Q*-criterion. The observation plane is provided in Figure 14i. In the rotating domain, two blades (blade number 1 and 2) are enlarged in (A), while the latter (blade number 2) is further enlarged in (B). The streamlines were drawn on an imaginary plane passing through the shaft centerline and the axial line on the shroud, and the plane was generated every 2° in the circumferential direction ($\theta$). The *Q*-criterion was indicated using a red color to visualize the vortex. The *Q*-criterion is one of the most popular vortex identification methods, proposed by Hunt et al. [43] in 1988. It is defined as the squared residual of the vorticity tensor norm subtracted from the squared strain-rate tensor norm under the Frobenius norm [44]. In general, the distribution of a vortex from the *Q*-criterion shows quite a similar pattern to the result of a visible test. The range of the *Q*-criterion in this study was $1.18543–1.18732 \times 10^6$ s$^{-2}$, which was accurately distributed in the center of flow separation, as shown in Figure 14. The additional descriptions for each feature are given once for the 15% set in Figure 14a, on behalf of all sets for different maximum thickness positions. The tip leakage vortex (TLV) occurred from the tip of blade number 1 with a thick and long distribution of the *Q*-criterion toward the next blade (blade number 2). This vortex flow, however, did not directly face the LE of blade number 2, but instead faced the pressure surface (PS). This pattern was similar for all sets.

In addition to the TLV, there was another vortex core parallel to the rotational direction, which is marked "recirculation". The recirculation can be specified with a strong backflow distributed around

the blade SS as the incidence angle (difference between the blade angle and flow angle) increases [45]. However, because the design specifications (flow rate, blade angle, diameter, etc.) were the same in this study, the difference in the incidence angle could not be defined with the variation of the maximum thickness position. Nevertheless, the recirculation tended to decrease gradually as the percentage of the maximum thickness position increased. This means that the maximum thickness position had a certain effect on the inlet flow mechanism. For each set in Figure 14, the white dotted line corresponding to the recirculation area of the 15% set is drawn in (B). Meanwhile, the tendency of the recirculation area for the maximum thickness position was still inversely proportional to the result of the main effect plot for the shroud span in Figure 11b. It can be seen that the strongest recirculation flow for the 15% set was the main cause for the remarkable decrease in efficiency (Figure 10); however, it was difficult to confirm whether the recirculation flow near the shroud affected the efficiency of the axial fan, at least for the sets from 20% to 35% (Figure 11b).

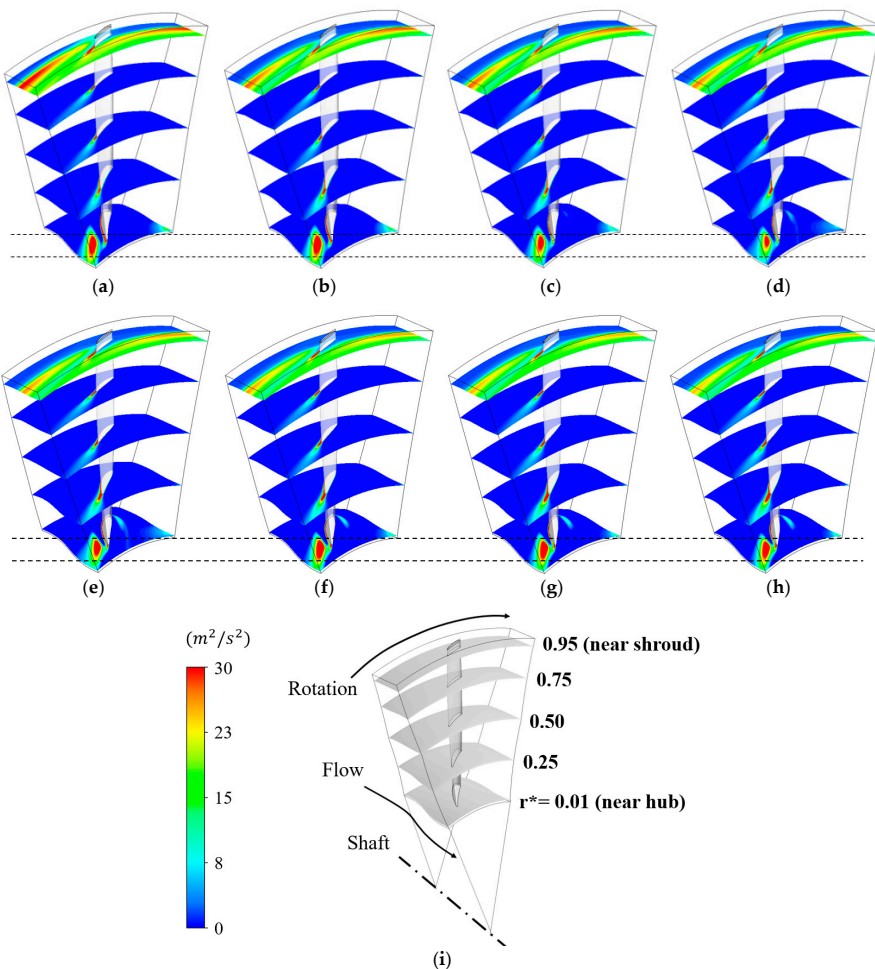

**Figure 12.** Distribution and range of turbulence kinetic energy on (**i**) the observation planes in a rotating domain for the (**a**) 15%, (**b**) 20%, (**c**) 25%, (**d**) 30%, (**e**) 35%, (**f**) 40%, (**g**), 45%, and (**h**) 50% sets ($\Phi/\Phi_{des} = 1.0$).

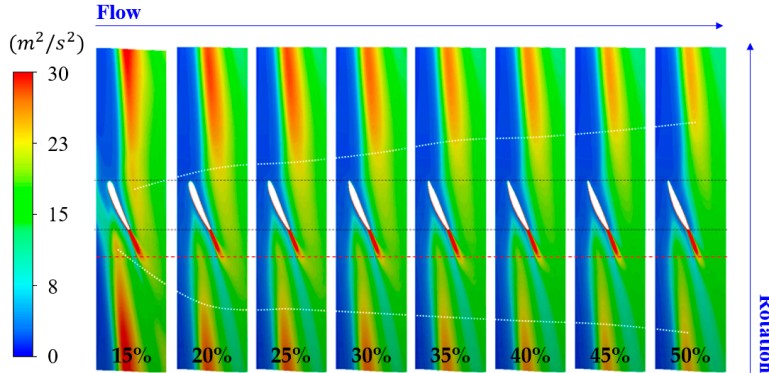

**Figure 13.** Distribution and range of turbulence kinetic energy near the shroud span with different maximum thickness positions ($\Phi/\Phi_{des} = 1.0$).

The trajectory of the TLF is further specified in Figure 15. The *x*- and *y*-axis of the graph are defined in Figure 16. The *x*-axis is the blade chord fraction ($\lambda$) with normalized values of 0 and 1 for the blade LE and TE at the hub span, respectively. The *y*-axis is a nondimensionalized axial distance as a function of the diameter of the axial fan. Each blade was drawn by the black curve following the camber (mean) line of the shroud span. The trajectory of the TLF was extracted from the center point of the vortex core, and converted to the actual length reflecting the circumferential length. It can be seen that the TLF from blade number 1 flowed to the PS of blade number 2. The life of the TLF could be classified into (A) inception, (B) transition, (C) stabilization, and (D) deformation, as indicated by the dotted black segments on the graph. Each segment was enlarged with the ignored scale ratio at the top of the graph. In (A), the TLF was developed and detached from the blade SS. It can be seen that a smaller percentage of the maximum thickness position led to faster detachment. In (B), the TLF away from the blade flowed with an unstable pattern with respect to the maximum thickness position. In (C), the TLF began to take a stable pattern with respect to the maximum thickness position. As the percentage of the maximum thickness position decreased, the trajectory of the TLF moved upstream. In (D), the trajectories were bent while maintaining the axial pattern of (C). This could be seen as the effect of blade number 2. An important conclusion from Figure 15 is that the trajectory of the TLF with respect to the maximum thickness position showed an axial pattern. In general, the TLF has a characteristic of being directed upstream as the flow rate decreased, i.e., as the incidence angle increases, and downstream as the incidence angle decreases [42]. The axial pattern confirmed in Figure 15 shows that the direction of TLF moved upstream (downstream) when the maximum thickness position was located near the LE (TE). As with the recirculation flow in Figure 14, it was not possible to define the variation in incidence angle in this study; however, the difference in maximum thickness position was considered to produce a similar effect to the change in incidence angle. It was also confirmed in Figure 9a that the slope tended to be horizontal at the low flow rate as the maximum thickness position moved toward the LE. All of these phenomena seemed that the incidence angle increased due to the axial component of the absolute velocity decreasing as the maximum thickness position moved toward the LE.

Figure 17 shows the contour of the circumferential velocity component ($V_{\theta}$) and the axial velocity component ($V_z$) on the meridional plane for tip clearance. In each legend, the circumferential velocity component is presented as a positive value (plus, "+") when it is opposite to the rotational direction of the axial fan blade, and the axial velocity component is presented as "+" when it coincides with the flow direction. All ticks in each contour have the same value as the label indicated for the 50% set. The nondimensionalized axial distance in the *x*-axis can be seen in Figure 16. From Figure 17a, the predominant flow in the circumferential direction through the tip clearance was from the PS to the SS. Moreover, as the maximum thickness position moved toward the TE, the distribution with the highest velocity gradually moved downstream. Since the area of the meridional plane for

tip clearance was the same for all sets, an increase in velocity resulted in an increase in flow rate. Therefore, the maximum thickness position seemed to affect the leakage loss from the tip clearance. The maximum thickness position and the location of the maximum leakage loss presented the same pattern. The backflow through the tip clearance can be confirmed from Figure 17b. All contours in Figure 17b had a negative (minus, "−") distribution. Thus, the predominant flow in the axial direction through the tip clearance was also from the PS to the SS. The distribution with the highest axial velocity component, however, moved upstream as the maximum thickness position moved toward the TE.

For the tip clearance, the area of the meridional plane ($A_t$), circumferential velocity component ($V_\theta$), axial velocity component ($V_z$), and TLF velocity component ($V_{TLF}$) are listed in Table 3. Each value of the velocity components was extracted as the averaged calculation at the meridional plane for tip clearance in Figure 17. The radial velocity component was excluded because its magnitude was too low to affect the results of the analysis. Despite the slight difference in the gradient, both circumferential and axial velocity components gradually increased as the maximum thickness position moved toward the TE, meaning the strongest and fastest countercurrent component being generated for the 15% set. As a result, the TLF velocity showed a pattern that gradually increased as the maximum thickness position moved toward the TE.

**Table 3.** Area and velocity components on the meridional plane for tip clearance ($\Phi/\Phi_{des} = 1.0$).

| Max. Thickness Position (%) | $A_t$ (m$^2$) | $V_\theta$ (m/s) | $V_z$ (m/s) | $V_{TLF}$ (m/s) |
|---|---|---|---|---|
| 15 | | 105.71 | −39.55 | 112.87 |
| 20 | | 106.52 | −39.59 | 113.64 |
| 25 | | 106.82 | −39.79 | 113.99 |
| 30 | 0.000436 | 106.71 | −39.48 | 113.78 |
| 35 | | 107.17 | −38.98 | 114.04 |
| 40 | | 107.92 | −38.90 | 114.72 |
| 45 | | 108.41 | −38.34 | 114.99 |
| 50 | | 108.82 | −37.60 | 115.13 |

The TLF rate ($\Phi_{TLF}$) through the tip clearance and the TLF angle ($\beta_{TLF}$) from Figure 16 are presented in Figure 18. The TLF rate is expressed as the flow coefficient from Equation (3). The TLF angle gradually decreased as the maximum thickness position moved toward the TE, i.e., the TLF moved downstream as the TLF angle decreased. This is consistent with the pattern of the TLF trajectory in Figure 15. Thus, it is understood that a more stable confluence with the main flow was achieved as the TLF trajectory moved downstream, leading to the intensity of the TKE decreasing in Figures 12 and 13. In addition, the TLF rate through the tip clearance, i.e., the leakage loss of the flow rate, increased as the maximum thickness position moved toward the TE. On the basis of the main effect plot for the shroud span in Figure 11b, it can be confirmed that a larger leakage loss resulted in a greater efficiency loss. The percentage in the bar graph in Figure 18 shows the ratio of the TLF rate divided by the design flow rate. It seems that the values were too low to have a direct effect on the performance of the axial fan; thus, the main effect plot for the shroud span in Figure 11b showed a low gradient. Nevertheless, it is necessary to pay attention to this value for the TLF rate when the tip clearance or the chord length at the blade shroud is increased. Meanwhile, it was difficult to say that the TKE became strong due to the TLF rate.

The detailed flow mechanism near the blade LE at $r^* = 0.95$–$1.00$ is presented in Figure 19a with the axial velocity contour and flow vector on the observation plane. The observation plane constitutes the red rectangular area identified with a yellow arrow in Figure 19b, which was one of the observation planes in Figure 14i. It was about 5% away from the blade LE at the shroud span. As denoted by the legend in Figure 19b, the contour became blue when the axial velocity component obtained a value of 0 or less. The blue contour region denotes backflow against the main flow direction, which is in good agreement with the distribution of the vectors. Meanwhile, each numbering was given for specific

ranges of the nondimensionalized axial distance. In accordance with Figure 14, the following footnotes could be attached to each range: region 1 (recirculation); region 2 (suction surface side); region 3 (blade and tip clearance); region 4 (pressure surface side); region 5 (TLF from the former blade). The white area occupied by the blade, region 3, became thinner as the maximum thickness position moved toward the TE.

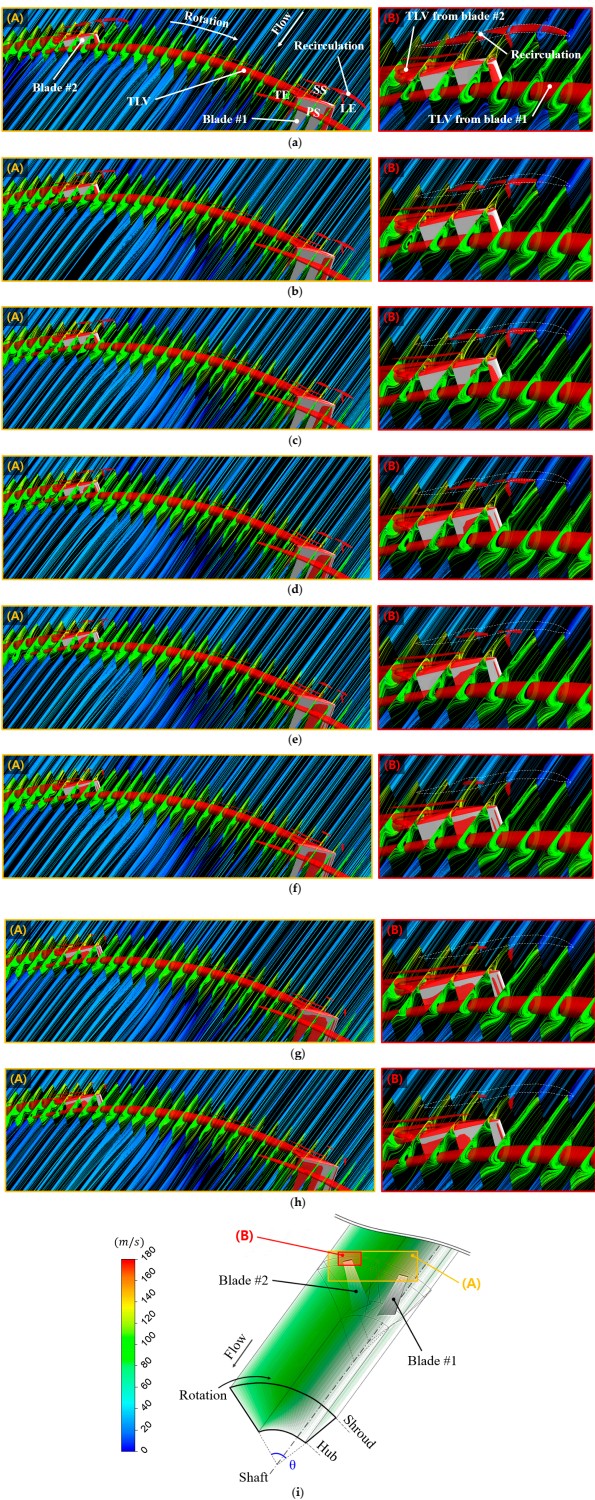

**Figure 14.** Internal flow characteristics with streamlines and $Q$-criterion on (**i**) the observation planes for the (**a**) 15%, (**b**) 20%, (**c**) 25%, (**d**) 30%, (**e**) 35%, (**f**) 40%, (**g**), 45%, and (**h**) 50% sets ($\Phi/\Phi_{des} = 1.0$).

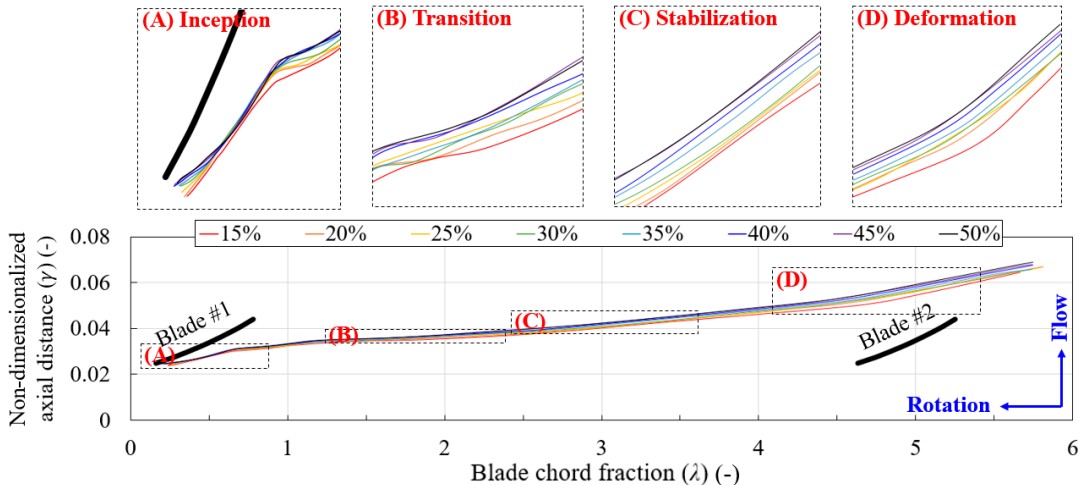

**Figure 15.** Trajectory of the tip leakage flow with different maximum thickness positions ($\Phi/\Phi_{des} = 1.0$).

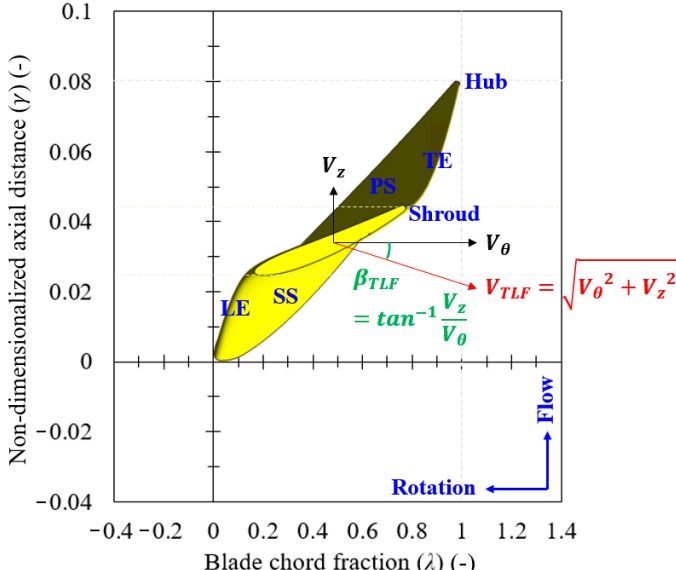

**Figure 16.** Expressions for the location of the blade and the velocity of tip leakage flow on a two-dimensional plane.

Most of all, the negative velocity contour was quite dominant. The backflow from region 4 was connected to the formation of recirculation in region 1. In accordance with Figure 14, the recirculation in region 1 gradually decreased as the maximum thickness position moved toward the TE. The most important point was the backflow in region 2, which also decreased gradually as the maximum thickness position moved toward the TE, showing the same tendency as region 1. In this study, as mentioned above, the inlet flow angle and the blade angle were the same for all sets. However, as shown in region 3 of Figure 19, the blade thickness fraction near the inlet was different for each set, such that the incoming flow with the same flow angle was in contact with the blade having a different inlet shape for each set. In other words, the flow was stagnated into a thicker solid as the maximum thickness position moved toward the LE. Accordingly, a stronger backflow was formed near region 2 (SS), and a faster flow was generated toward the positive streamwise direction near region 4 (PS). As a result, the strong backflow in region 2 affected the growth of recirculation in region 1, which could be treated as the mechanism of the flow angle. Meanwhile, in region 5, the eye of the TLF gradually

moved downstream as the maximum thickness position moved toward the TE. This is consistent with the results confirmed in Figures 13 and 15.

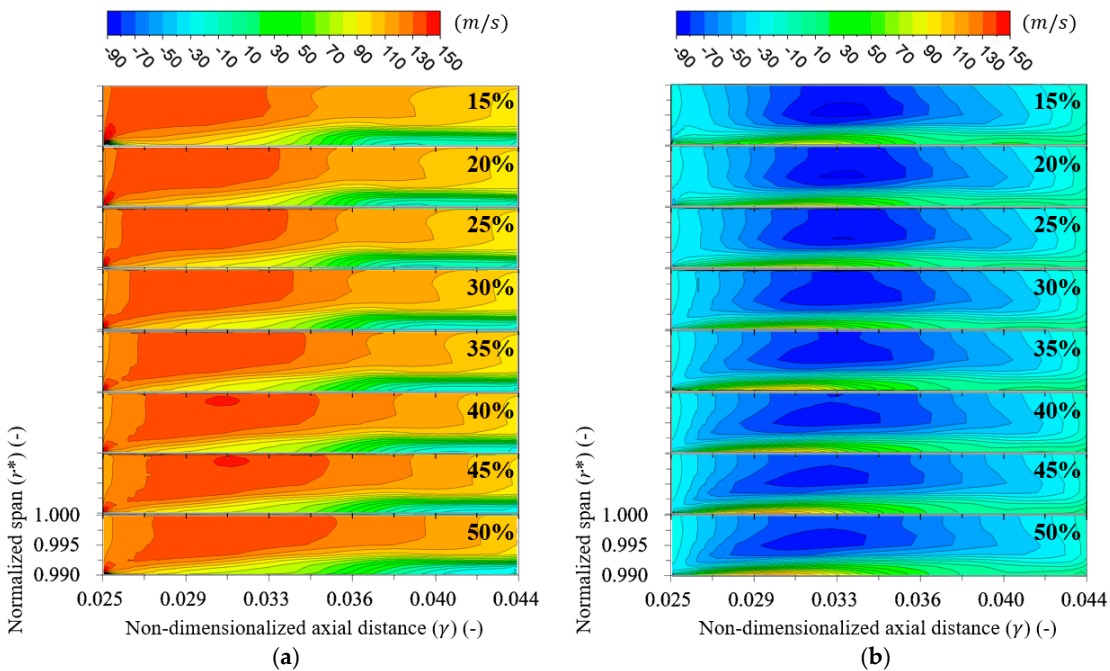

**Figure 17.** Contour of (**a**) the circumferential velocity component ($V_\theta$) and (**b**) the axial velocity component ($V_z$) on the meridional plane of the tip clearance with different maximum thickness positions ($\Phi/\Phi_{des} = 1.0$).

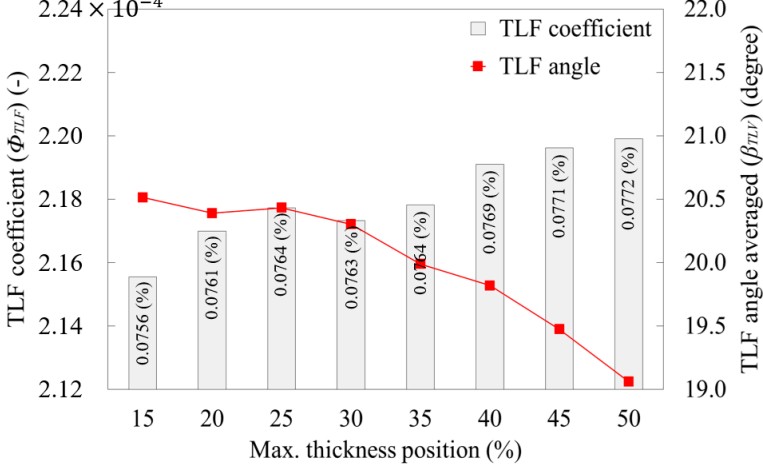

**Figure 18.** Flow rate and flow angle through the tip clearance with different maximum thickness positions ($\Phi/\Phi_{des} = 1.0$).

Figure 20 presents the pressure rise curves corresponding to the nondimensionalized axial distance, $\gamma$. There was a remarkable pressure loss in the range $\gamma \approx 0.005$–$0.020$. This range was located at the same position as the area of recirculation and backflow identified in Figure 19. The highest pressure loss was observed for the 15% set, where the recirculation and backflow were strongest. The pressure loss tended to decrease as the maximum thickness position moved toward the TE. Thus, it can be seen that the pressure rise was negatively affected by the recirculation and backflow near the blade inlet ($\gamma \approx 0.005$–$0.020$). However, the 15% set showed the most superior pressure rise during the operation.

In the case of the 50% set, despite the lowest negative effect near the inlet, the pressure rise from the blade was relatively insufficient, leading to the worst ranking among the eight sets.

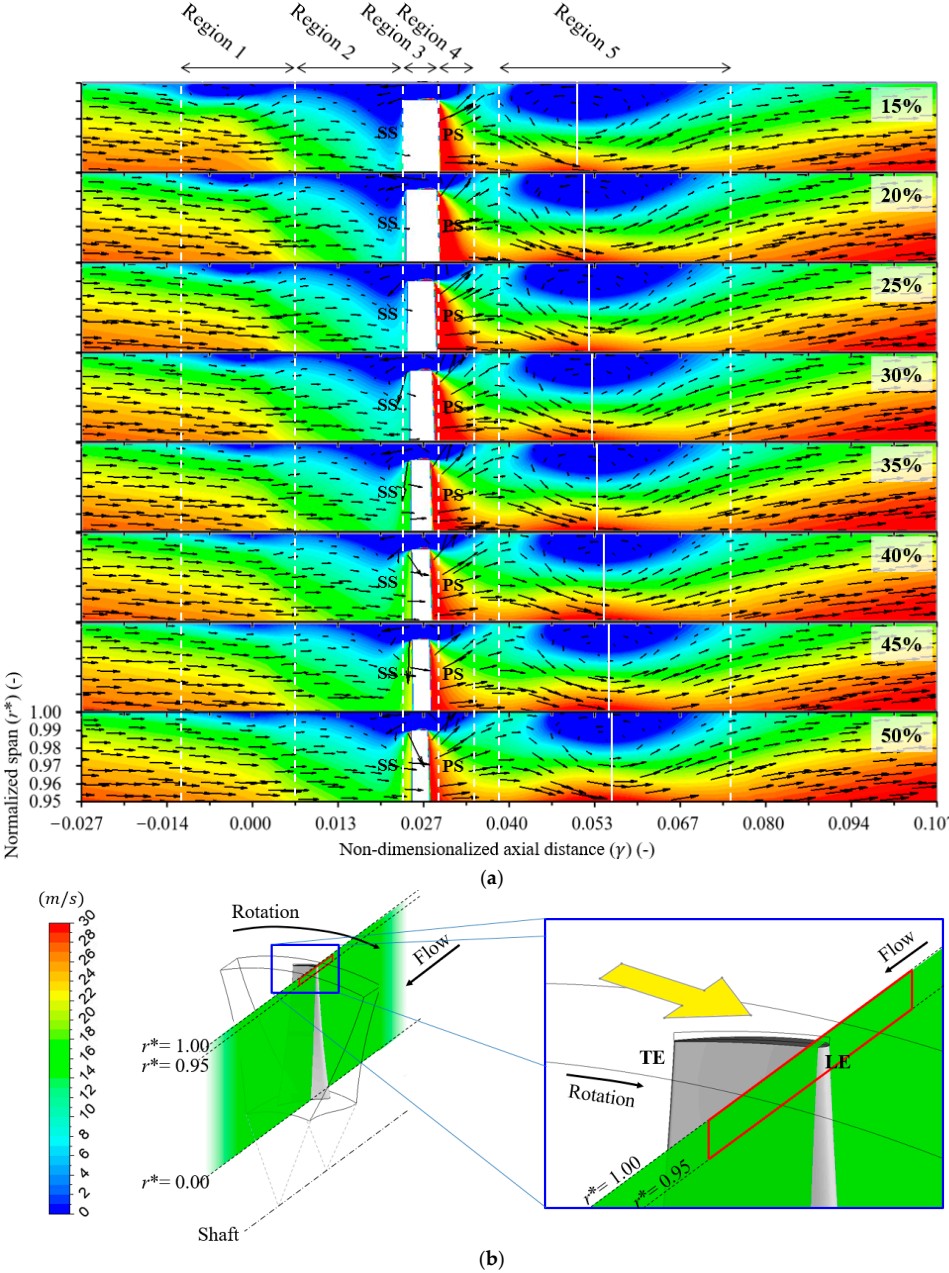

**Figure 19.** (**a**) Axial velocity contour and flow vector with different maximum thickness positions on (**b**) the observation plane ($\Phi/\Phi_{des} = 1.0$).

The blade loading distribution is shown in Figure 21. The *y*-axis of the graph constitutes the normalized streamwise position from the LE (0) to TE (1). The blade loading was greater near the shroud span in Figure 21b than the hub span in Figure 21a. In Figure 21a, different pressure drops were observed near the blade LE (streamwise position $\approx$ 0.0–0.3). On the PS, a smaller percentage of the maximum thickness position resulted in a deeper pressure drop. On the contrary, on the SS, the pressure increased as the percentage of the maximum thickness position increased. This tendency could be interpreted as the geometry near the blade LE having an impact on the blade loading. In Figure 21b, for the shroud span, this tendency near the blade LE was maintained; however, the magnitude of the difference was greater with each set. In particular, as can be seen from the enlarged circle in Figure 21b,

an additional pressure drop was observed on the SS near the blade TE for certain sets, which was not observed in the hub span. The pressure drop was more severe as the percentage of the maximum thickness position decreased. Such an uneven pressure (blade loading) distribution resulted from the increase in incidence angle for some cases [10], including the noise or structural stability of the blade.

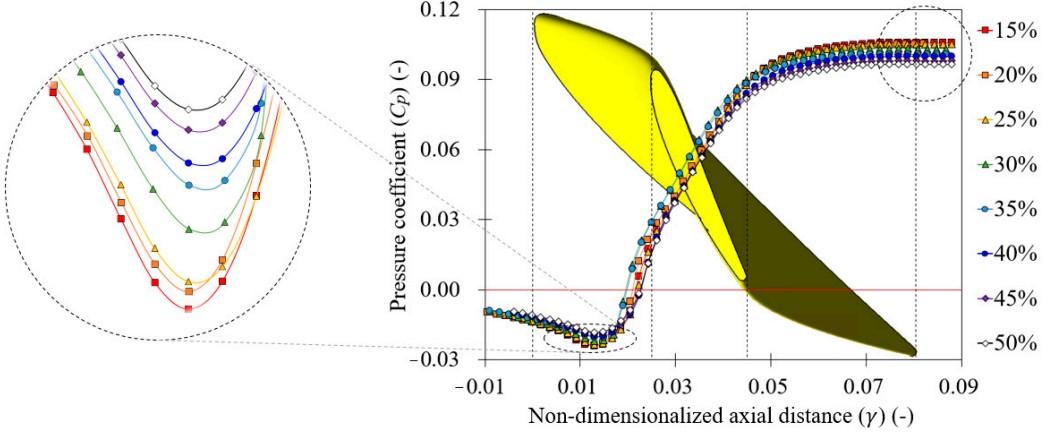

**Figure 20.** Pressure rise curves with different maximum thickness positions ($\Phi/\Phi_{des} = 1.0$).

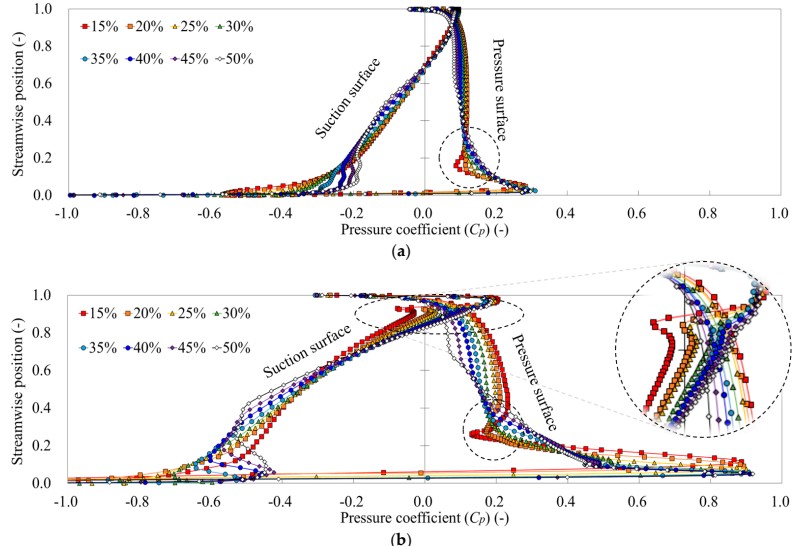

**Figure 21.** Blade loading curves with different maximum thickness positions ($\Phi/\Phi_{des} = 1.0$): (**a**) hub span; (**b**) shroud span.

### 4.4. Trailing Edge Treatment Effect

An additional thickness treatment was implemented to potentially recover the uneven pressure distribution and efficiency dissipation identified in the above sections. Fukano et al. [46] reported that vortex-induced noise and vibration can be minimized with a thickness treatment at the blade TE, because of the decrease in the wake and turbulence boundary layer. For this section, the thickness of the blade TE was reduced from the 15% set for the maximum thickness position, such that a total of three sets were designed. The other design parameters and specifications were unaltered. The performance with respect to the thickness treatment near the TE is shown in Figure 22. As the thickness was reduced for the 3 mm set, both total pressure and efficiency tended to increase. Compared to the performance characteristics with respect to the maximum thickness position in Figure 10, it seems that the total pressure and efficiency were more sensitive depending on the thickness treatment near the TE than the maximum thickness position.

Figure 23 shows the pressure rise distribution, which is in contrast to the results in Figure 20. Since there was no geometrical change near the blade LE, it was difficult to expect any variation in the flow mechanism near the inlet. Therefore, the portion indicated by the dotted circle showed almost the same distribution despite the change in thickness near the TE. It could also be predicted that the distribution of recirculation and backflow near the blade LE at the shroud span remained for all three sets, as observed in Figure 14a. However, the 1 mm set showed the greatest pressure rise when the flow passed through the blade. As a result, a thinner thickness near the TE resulted in a higher pressure rise.

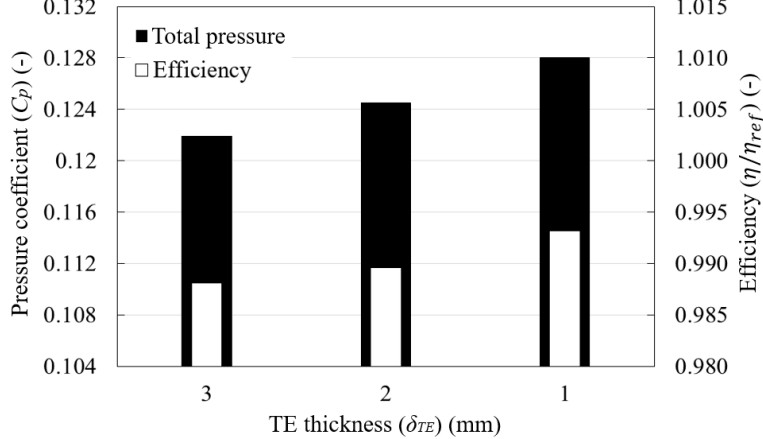

**Figure 22.** Performance with different trailing edge thickness treatments ($\Phi/\Phi_{des} = 1.0$).

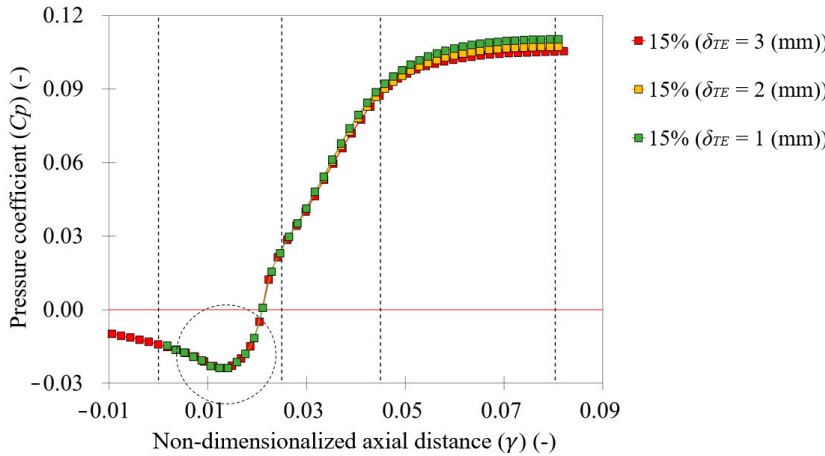

**Figure 23.** Pressure rise curves with different trailing edge thickness treatments ($\Phi/\Phi_{des} = 1.0$).

Figure 24 shows the blade loading distribution, which is in contrast to the results in Figure 21. As expected, the blade loading distribution near the LE was not correlated with the TE thickness treatment. However, especially in the shroud span, the nonuniform pressure distribution on the SS near the blade TE tended to be significantly improved as the TE thickness decreased. As a result, the axial fan obtained a more stable loading distribution near the TE as its thickness decreased. It is advantageous to design as thin a TE as possible if the manufacturing process does not suffer from cost restraints.

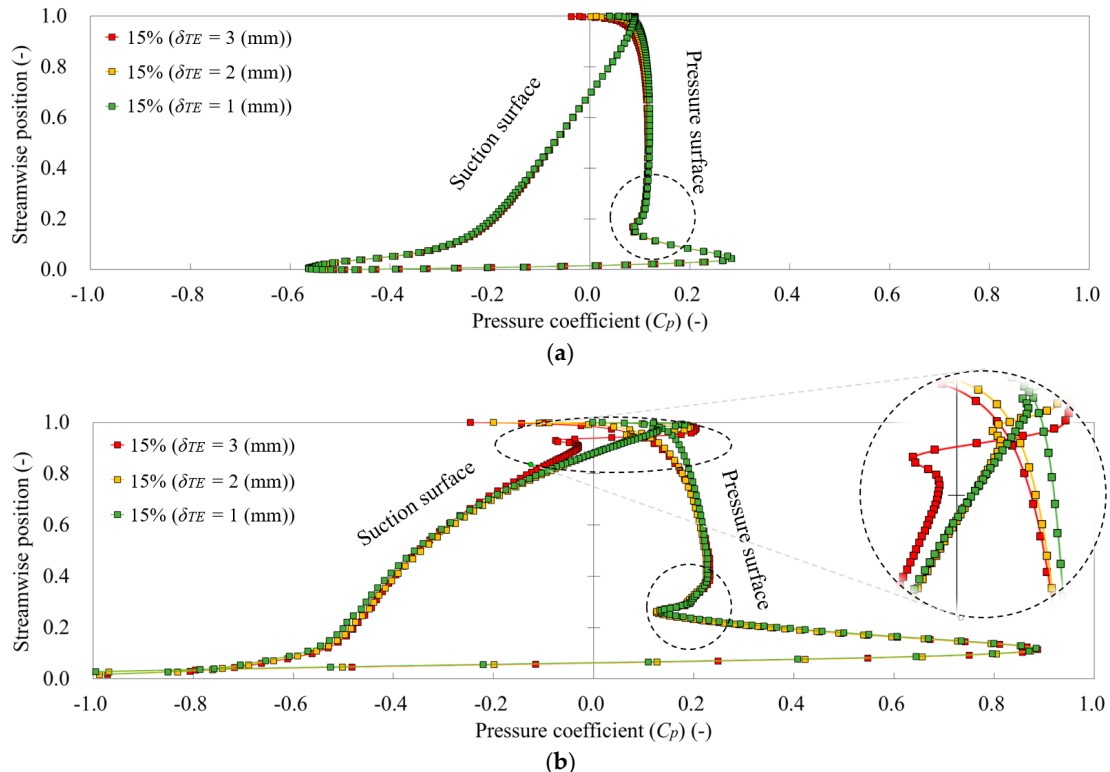

**Figure 24.** Blade loading curves with different trailing edge thickness treatments ($\Phi/\Phi_{des} = 1.0$): (**a**) hub span; (**b**) shroud span.

## 5. Conclusions

In this study, the effect of the maximum thickness position of an airfoil on the performance and aerodynamic characteristics of an axial fan was examined using numerical analysis. The effect of the maximum thickness position was discussed from various perspectives. Moreover, an additional discussion related to the TE thickness treatment was presented as a solution for the unstable state. The results of this study can be summarized as follows:

1.  The maximum thickness position affected the performance of the axial fan. Performance curves for both the total pressure and the efficiency showed up–down shifts with almost the same slope. The total pressure rise decreased as the maximum thickness position moved toward the TE, including the off-design flow rates. The efficiency was predicted to be generally high for the 30% and 35% sets. Left–right shifts of the BEP were not observed.
2.  At the design flow rate, the total pressure decreased as the maximum thickness position moved toward the TE. In terms of efficiency, the highest performance could be expected when the maximum thickness position was at 30%, as recommended in the empirical equation.
3.  According to the main effect plot for each span, the total pressure tended to decrease in both the hub and the shroud span as the maximum thickness position moved toward the TE. In terms of efficiency, the highest efficiency was expected at the maximum thickness position of 30% in the hub span; however, in the shroud span, the efficiency tended to decrease as the maximum thickness position moved toward the TE. The main effect for the efficiency near the shroud span had an inversely proportional correlation with the TLF rate, describing the leakage loss.
4.  The intensity of the TKE of the TLF strengthened as the maximum thickness position moved toward the LE. The trajectory of the TLF moved upstream (downstream) when the maximum thickness position was located near the LE (TE). The intensity of the TKE was determined from

the direction of the TLF. However, it was difficult to establish a correlation between the TLF rate and the TKE.

5. The maximum thickness position affected the growth of the recirculation area near the blade SS. The recirculation area tended to gradually increase as the maximum thickness position moved toward the LE because of a stronger backflow near the LE. The recirculation and backflow led to pressure loss near the LE.

6. The nonuniform blade loading distribution on the PS near the blade LE and the SS near the blade TE was more severe as the maximum thickness position moved toward the LE.

7. As the thickness near the TE was reduced, both the total pressure and the efficiency tended to increase. The pressure rise and the blade loading near the LE were not affected depending on the thickness treatment near the TE. Particularly in the shroud span, the nonuniform blade loading distribution on the SS near the blade TE was improved significantly as the thickness decreased.

**Author Contributions:** Conceptualization, Y.-S.C.; formal analysis, Y.-I.K. and Y.-S.C.; funding acquisition, K.-Y.L. and S.-H.Y.; investigation, Y.-I.K. and Y.-S.C.; methodology, Y.-S.C.; project administration, S.-H.Y.; resources, S.-Y.L. and K.-Y.L.; software, K.-Y.L.; supervision, Y.-S.C.; validation, Y.-I.K. and S.-Y.L.; writing—original draft, Y.-I.K. and Y.-S.C.; writing—review and editing, Y.-I.K. and Y.-S.C. All authors read and agreed to the published version of the manuscript.

**Funding:** This research was conducted under grant No. 20172010106010 from the Korea Institute of Energy Technology Evaluation and Planning (KETEP). The authors gratefully acknowledge this support.

**Conflicts of Interest:** The authors declare no conflict of interest.

## Nomenclature

| | |
|---|---|
| BEP | best efficiency point |
| DAQ | data acquisition |
| DOE | design of experiments |
| KITECH | Korea Institute of Industrial Technology |
| LE | leading edge |
| NACA | National Advisory Committee for Aeronautics |
| PS | pressure surface |
| RANS | Reynolds-averaged Navier–Stokes |
| RMS | root mean square |
| RSM | response surface methodology |
| SS | suction surface |
| SST | shear stress transport |
| TE | trailing edge |
| TKE | turbulence kinetic energy |
| TLF | tip leakage flow |
| TLV | tip leakage vortex |
| $A$ | area |
| $C_{m2}$ | meridional component of absolute velocity at the blade outlet |
| $C_p$ | pressure coefficient |
| $C_\mu$ | nondimensional model constant (0.09) |
| $c$ | chord length |
| $F_i$ | body force |
| $k$ | turbulence kinetic energy |
| $l$ | turbulence length scale |
| $N$ | rotational speed |
| $N_s$ | specific speed |
| $P_t$ | total pressure |
| $Q$ | volume flow rate |

| | |
|---|---|
| $r^*$ | normalized span |
| $r_h$ | radius of hub |
| $r_s$ | radius of shroud |
| $s$ | blade pitch |
| $T_u$ | turbulence intensity |
| $U_2$ | circumferential component of rotational velocity at the blade outlet (tip speed) |
| $V_{TLF}$ | velocity of tip leakage flow |
| $V_z$ | axial velocity component |
| $V_\theta$ | circumferential velocity component |
| $W_1$ | relative velocity of air at the blade inlet |
| $\beta_{TLF}$ | tip leakage flow angle |
| $\gamma$ | nondimensionalized axial distance |
| $\delta$ | thickness |
| $\delta_m$ | maximum thickness |
| $\delta_t$ | tip clearance |
| $\eta$ | efficiency |
| $\lambda$ | blade chord fraction |
| $\mu$ | viscosity coefficient |
| $\nu$ | kinematic viscosity coefficient |
| $\rho$ | density |
| $\Phi$ | flow coefficient |
| $\Phi_{TLF}$ | tip leakage flow coefficient |
| $\tau_{ij}$ | viscous stress tensor |
| $\omega$ | turbulence eddy frequency |
| $\omega_N$ | angular velocity |

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
