# Peer review of "Numerical Investigation of Performance and Flow Characteristics of a Tunnel Ventilation Axial Fan with Thickness Profile Treatments of NACA Airfoil"

_energies, doi:10.3390/en13215831_

Round 1

Reviewer 1 Report

The authors have chosen a interesting topic. Although tip leakage flow and tip vortex aerodynamics is an rather old filed of research, it is still not fully answerred. Especially new experimental and numerical methods do offer new scientific results.

In the paper, signifikant effort was put into the TLF and TKE analysis - which is ok from my point of view.

The weak points I would like to have improved are the following:

a) I do not see the strong linkage between experiments and numerics - other than using the same geometries. There is no direct comparision of exprimental and numerical performance of the fan for the different thickness positions. Fig. 7 shows just one comparision and this is by far to coarse. The authors should in detail show the range of flow coeff. between 0,8 and 1,2 - the rest is simply out of interest.

b) The TKE-analysis in the second part of the paper is not supported by any experimental data - I am wondering why this is the case. A good valiation of the local flow field for at least on geometry would create the soundness and reliability which is needed to follow the entire second part of the paper.

c) Uncertainty analysis is missing - at least for the experiments shown

d) Our own experience shows that the difference between the exp. setup (e.g. inlet bellmouth plus inlet hub cover) vs. the numerial setup (e.g. full cyclindrical part without bellmouth and cover) is creating different behaviour - especially with regards toip vortex behaviour. This should be discussed and answerred

e) I am missing any information about the turbulence-related setup in the numerics, e.g. what is the inlet Tu-length scale, what is the inlet Tu-level and how does it change (decay) from the do,ain inlet to the fan - and is this in line with the experimental conditions? What feature for the turbulence model were used. Is the calculation with or without transition?

f) in Figure  14 the areas A, B, C, D should be clearly marked in the lower part.

Reviewer 2 Report

The authors aim to assess the effect of the profile thickness of an airfoil on performance and aerodynamic characteristics of the axial fan using numerical simulations.

The study is interesting, and the topic is important. Moreover, the study is very well presented. However, there are minor issues, which needs an additional attention to improve the quality of the paper.

Although the abstract is quite informative, it could be more focused on the novelty aspect.  In current form it serves a role of introduction, which is not recommended. The authors provide details on the results instead of pointing the main findings and explaining how they filled the knowledge gaps.

The introduction could be also improved. The discussion on some of the references is too short. Beside pointing what other researchers did in the past the authors should also provide main findings. The literature review should guide the reader to the knowledge gap, which is partially achieved, but several references are treated superficially.

In the numerical setup the authors do not discuss the issue of Mach number and compressibility. In the results this aspect is also omitted. Could the authors please comment on that? The numerical model is discussed in not enough detail. The authors should include main equations into the model description.

The very good aspect of the study is a validation of numerical results against experimental data, which showed good agreement in terms of considered parameter.

minor comments:

  • English needs improvement. there are clumsy and incorrect expressions such as: “the thickness profile of the airfoil”
  • “energy performance” better “efficiency”
  • “Understanding the mechanism…” what kind of mechanism?
  • instead of horsepower use kW
  • List of symbols is missing

Reviewer 3 Report

please, see attached file.

Reviewer 4 Report

This paper is aiming to evaluate the effect of the thickness profile of airfoil on the energy performance and aerodynamic characteristics of the axial fan through numerical analysis.I consider that this paper requires a major revision by the authors, that shall consider the following comments:

  1. In the section 1 - Introduction, it is said: “The effect of the maximum thickness dimension of airfoil (referred to as the last two codes of NACA 4-digit) on the performance of the axial fan was studied by Sarraf et al. [20], however, papers and reports on thickness profile are still insufficient.” A summary of the work carried out by Sarraf et al. [20] shall be presented in order to justify the need of the work presented in this paper.
  2. In the section 3. Numerical analysis setup it is not justified that the mesh used in this study has the adequate refinement. A study of the mesh should be presented.
  3. In the section 4.1. Validation for numerical analysis, the temperature of the air and the atmosférica pressure during validation tests shall be reported.

Round 2

Reviewer 1 Report

Thank you for the amendments and clarifications in the paper.

Although I am not 100% convinced with regards to the quality of comparison between exp. and CFD as well as the uncertainty analysis (which is still not shown as an analysis inclusing bais and systematic error), I see that the quality of the paper has been improved.

I still recommend - as before - to change Fig. 8 or add a Fig.8b in oder to show a zoom-area betwn phi=0,8 and 1,2, where Exp. and CFD data is available - this also helps to include the uncertainty information.

Author Response

Please see the attachment. Thanks for the comments.

Reviewer 2 Report

The authors have addressed all my comments and revised the manuscript accordingly.

Therefore, my recommendation is to accept the manuscript in the present form.

Author Response

Thanks for the recommendation.

Reviewer 4 Report

0

Author Response

Thanks for the recommendation.